# Identification of motor neurons and a mechanosensitive sensory neuron in the defecation circuitry of *Drosophila* larvae

**Wei Zhang[1,2†], Zhiqiang Yan[1,2,3,4†], Bingxue Li[3,4], Lily Yeh Jan[1,2], Yuh Nung Jan[1,2*]**

[1]Department of Physiology, Howard Hughes Medical Institute, University of California, San Francisco, San Francisco, United States; [2]Department of Biochemistry and Biophysics, University of California, San Francisco, San Francisco, United States; [3]State Key Laboratory of Genetic Engineering, Fudan University, Shanghai, China; [4]Collaborative Innovation Center for Genetics and Development, Fudan University, Shanghai, China

**Abstract** Defecation allows the body to eliminate waste, an essential step in food processing for animal survival. In contrast to the extensive studies of feeding, its obligate counterpart, defecation, has received much less attention until recently. In this study, we report our characterizations of the defecation behavior of *Drosophila* larvae and its neural basis. *Drosophila* larvae display defecation cycles of stereotypic frequency, involving sequential contraction of hindgut and anal sphincter. The defecation behavior requires two groups of motor neurons that innervate hindgut and anal sphincter, respectively, and can excite gut muscles directly. These two groups of motor neurons fire sequentially with the same periodicity as the defecation behavior, as revealed by in vivo Ca$^{2+}$ imaging. Moreover, we identified a single mechanosensitive sensory neuron that innervates the anal slit and senses the opening of the intestine terminus. This anus sensory neuron relies on the TRP channel NOMPC but not on INACTIVE, NANCHUNG, or PIEZO for mechanotransduction.

**\*For correspondence:** yuhnung.
jan@ucsf.edu

†These authors contributed
equally to this work

**Competing interests:** The
authors declare that no
competing interests exist.

**Reviewing editor**: K
VijayRaghavan, National Centre
for Biological Sciences, Tata
Institute of Fundamental
Research, India

## Introduction

Defecation is important for food processing that provides nourishment to the animal. It eliminates waste (feces) from the digestive tract via the anus (*Thomas, 1990*; *Heaton et al., 1992*; *Lembo and Camilleri, 2003*), an unglamorous but essential body function. Compared to the extensively studied feeding behavior, defecation has received relatively little attention. Malfunction of defecation can lead to constipation and other diseases (*Lembo and Camilleri, 2003*), and abnormal development of neural circuits governing defecation may underlie birth defects such as Hirschsprung's disease due to elimination of intestinal ganglion cells required for bowel peristalsis (*Romeo et al., 1994*; *Passarge, 2002*), one of the major birth defects of the digestive system afflicting one in 4000 of the population.

*Drosophila* larvae provide a useful model system for the studies of feeding behavior and nutrition intake (*Ikeya et al., 2002*; *Rulifson et al., 2002*; *Hwangbo et al., 2004*; *Bader et al., 2007*). With an array of feeding assays and powerful genetic tools, these animals have yielded valuable information regarding the basis of the feeding behavior (*Shen, 2012*; *Zhao and Campos, 2012*; *Bhatt and Neckameyer, 2013*). However, modulation of defecation behaviors has received much less attention until recently (*Edgecomb et al., 1994*; *Cognigni et al., 2011*). Harnessing the experimental resources of this model system for the study of gut movements and the underlying neural basis should also help us understand the mechanisms of the defecation behavior.

In the larval intestines, peristaltic movements of the digestive tract push food from the anterior towards the posterior end. The rate of flow depends on various signals from gut cells and associated

**eLife digest** In animals, the final stage in the digestion of food is the removal of waste from the body. Until recently, however, defecation has received less attention than other aspects of digestion such as feeding behavior and nutrition.

Fruit flies, also known as *Drosophila,* are commonly used in research as a model of animal biology. Food is moved through the digestive tract of fruit fly larvae when the muscles that circle the wall of the intestine contract. This process continues until the waste reaches the anus and is expelled from the body.

Now Zhang et al. have found that when fruit fly larvae defecate, the muscles at the end of the intestine contract just before the muscles in the anus contract. The nervous system controls these muscles via sequential firing of two sets of nerve cells that connect to the intestine and anus muscles, respectively.

Zhang et al. also identified a nerve cell that can sense when the anus is opened and relay this information back to the nervous system. The nerve cell is activated when stretched by the opening of the anus in a process that requires a protein called NOMPC.

Problems with defecation can lead to constipation and other diseases. For example, Hirschsprung's disease—a birth defect that affects one in 4000—is caused by abnormal development of the nerve cells that control muscles in the gut. Experiments on fruit flies could help us to understand how defecation works in humans and to develop new treatments for disease.

neurons (*Benoit and Tracy, 2008*; *Schoofs et al., 2009*). In *Caenorhabditis elegans* two groups of excitatory GABAergic motor neurons have been identified with partially redundant functions in activating enteric muscle cells (EMCs) (*McIntire et al., 1993*). Little is known about the motor control of gut movements in *Drosophila* larvae or any involvement of sensory neurons for defecation.

Mechanosensation is essential for many activities of *Drosophila*. Studies in adult flies have demonstrated that internal sensory neurons are important in regulating behaviors such as feeding, defecation, and egg laying (*Yang et al., 2009*). Whereas recent studies have identified mechanosensitive channels in specific sensory neurons in the larval body wall for harsh or gentle touch (*Kim et al., 2012*; *Yan et al., 2013*), whether and how a larva senses stretches of its internal organs is unknown nor have the neurons and channels mediating such mechanosensation been identified.

In this study we establish *Drosophila* larvae as a model system to study defecation behavior by performing studies of larvae 96 hr after egg laying (AEL). First, we show that *Drosophila* larvae exhibit rhythmic cycles of sequential contractions of the hindgut and the anal sphincter to expel feces. Second, we identify the motor neurons that innervate the hindgut and anal sphincter and show that these two groups of neurons fire sequentially with the same periodicity as the defecation cycle. Unexpectedly, we found that a single sensory neuron innervates the anal slit to sense its opening. Finally, we show that the TRP channel NOMPC but not other known mechanosensitive channels in *Drosophila* is required for the mechanosensation of this anus sensory neuron.

## Results

### Sequential contractions and innervations of the hindgut and anus sphincter

The *Drosophila* larval hindgut is the last part of the intestine, posterior to the Malpighian tubule, on the dorsal side under larval cuticle. At the posterior end of the hindgut is anal sphincter, which has a layer of thick sphincter muscles and a much narrower canal (*Figure 1A*, *Figure1—figure supplement 1*) (*Murakami and Shiotsuki, 2001*). Because the *Drosophila* larval body wall is transparent, contractions of the hindgut and anal sphincter can be monitored in vivo. Fluorescent markers, expressed with a hindgut-specific byn-Gal4 (*Johansen et al., 2003*), allowed visualization of contractions of the larval hindgut and anus sphincter in whole-mount of living larvae (*Figure 1B*). The defecation behavior consists of sequential contractions of the posterior hindgut and anal sphincter in a very stereotypical manner (*Figure 1B,D*), leading to opening of the anal slit to expel feces out of the lumen. This defecation process is repeated every 38 s at 25°C (*Figure 1D*). To demonstrate those gut movements

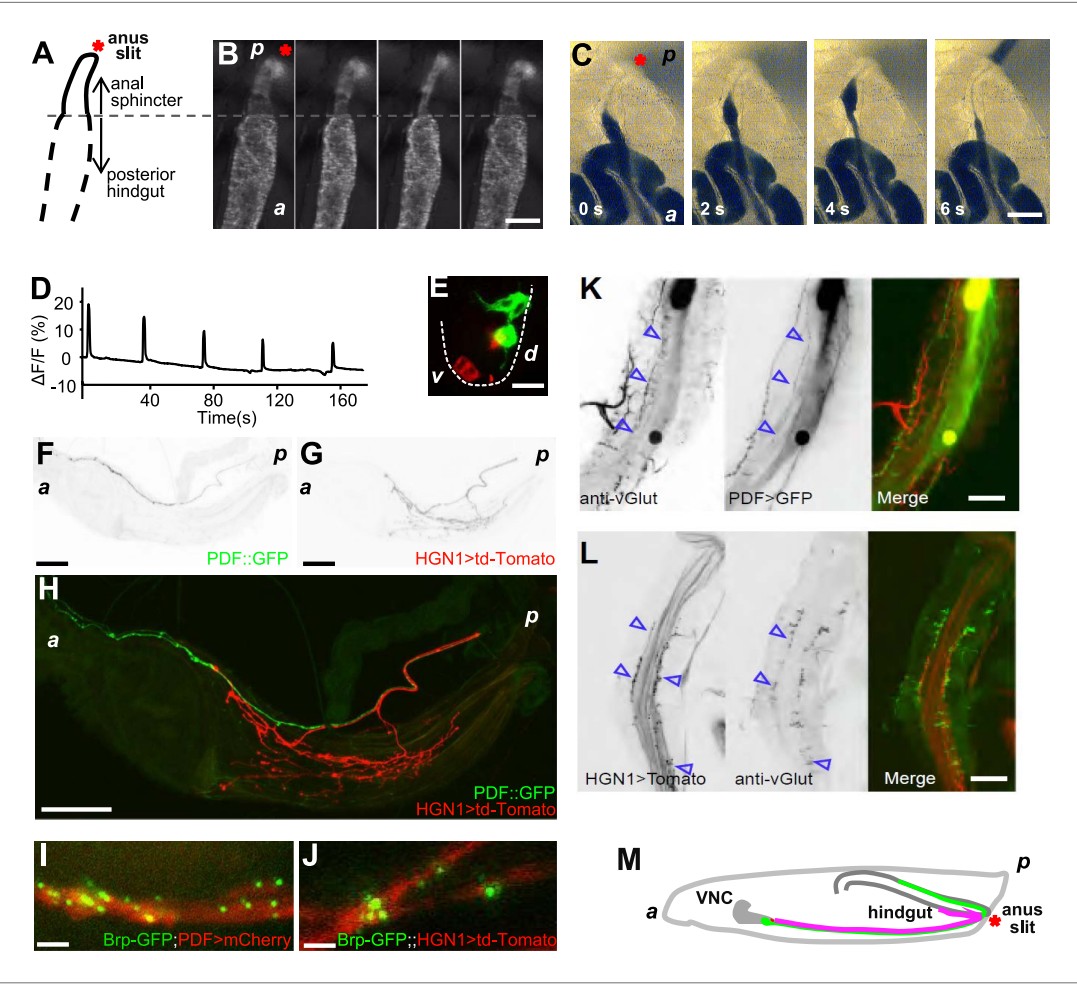

Figure 1. The periodical defecation process of the *Drosophila* larvae and the innervation of hindgut and anal sphincter by motor neurons. (**A** and **B**) Sequential contractions of the posterior hindgut and anal sphincter. (**A**) Schematic representation of the posterior hindgut and anal sphincter. The posterior hindgut: dashed line; anal sphincter: solid line. (**B**) Posterior hindgut and anal sphincter contract sequentially in the defecation process (visualized with byn-Gal4 > GFP). From left to right: quiescent state, contraction of the posterior hindgut, contraction of the anal sphincter, and back to resting state (scale bar: 100 µm). (**C**) Posterior hindgut and anal sphincter contract sequentially to expel the feces out. From left to right: quiescent state, feces pushed to anus via hindgut contraction, contraction of anal sphincter, and end of defecation cycle (scale bar: 100 µm). (**D**) Time course of defecation cycle measured by GFP fluorescence intensity as in (**B**). A region of interest (ROI) was drawn on the posterior hindgut; the fluorescence intensity in ROI increased due to the tissue compression. (**E**) Cell bodies of PDF neurons (green, PDF-GFP) and HGN1 neurons (red, HGN1 > UAS-tdTomato) in the terminal segment of the ventral nerve cord (VNC) (scale bar: 20 µm). (**F**–**H**) Innervations in the gut of neurons labeled with PDF (green, PDF-GFP) and HGN1 (red, HGN1 > UAS-tdTomato). PDF (**F**) and HGN1 (**G**) axons (scale bar: 100 µm). (**I** and **J**) Buttons of PDF (**I**) and HGN1 (**J**) neurons labeled by Brp-GFP (green dots) along axons (red) (scale bar: 10 µm). (**K** and **L**) Anti-vGlut staining of PDF and HGN1 axons on hindgut. (**K**) Anti-vGlut staining on hindgut (red) in PDF-GFP larvae (green). Blue arrow heads indicate axon terminals (scale bar: 50 µm). (**L**) Anti-vGlut staining on anal sphincter (green) in HGN1 > tdTomato larvae (red). Blue arrow heads indicate axon terminals (scale bar: 50 µm). (**M**) Schematic representation of the PDF and HGN1 neurons and their innervations on gut in a whole animal lateral view.

The following figure supplements are available for figure 1:

**Figure supplement 1**. The muscle structures of hindgut and anal sphincter.

**Figure supplement 2**. Glutamatergic innervations of motor neurons on the hindgut.

triggered defecation, we fed the larvae with yeast laced with blue food dye and video taped their defecation cycle. As shown in *Figure 1C* and *Video 1*, each sequential contraction of hindgut and anus sphincter triggered a defecation cycle to expel feces out of the body.

To investigate the neural basis for the gut movements, we searched for the neuronal innervation of the hindgut and anal sphincter muscles. Since the axons that innervate the hindgut are from the most posterior pair of the axon bundles in the ventral nerve cord (VNC), the cell bodies of the neurons that innervate the hindgut and anal sphincter are most likely in the terminal segments of the VNC. We identified two groups of neurons, labeled by PDF-Gal4 and HGN1-Gal4 (*Nassel et al., 1993*; *Edgecomb et al., 1994*; *Renn et al., 1999*; *Landgraf et al., 2003*; *Cognigni et al., 2011*), which innervate the posterior hindgut and anal sphincter, respectively (*Figure 1E–G*). These neurons have their cell bodies in the terminal segments of VNC (*Figure 1E*) and send their axons along the midline of the ventral body wall to the posterior end of the larva, where they enter the hindgut. Within the hindgut, the HGN1 axons extend posteriorly to the anus sphincter surface to form dense arborizations over the muscles, while the PDF axons arborize over the posterior two-third of the hindgut with refined branches (*Figure 1F–H*). The PDF and HGN1 neurons are glutamatergic, as they could be labeled with antibody staining against *Drosophila* vGlut (*Daniels et al., 2008*) (*Figure 1K,L*). The axonal branches of PDF neurons on the hindgut can also be labeled with vGlut-Gal4 (*Mahr and Aberle, 2006*) (*Figure 1—figure supplement 2*), indicating that they are likely glutamatergic motor neurons. The labeled neurons in both cases have their axon terminals in close proximity of the gut muscles and form abundant bouton structures (*Figure 1I,J*). These results suggest that PDF and HGN1 neurons, which are likely motor neurons, might play a role in regulating hindgut contractions (*Figure 1M*).

## Gut muscles receive excitatory input from PDF and HGN1 neurons

In order to explore the functional connection between HGN1 neurons and anal sphincter muscles, we expressed Channlerhodopsin-2 (ChR2), a light activated cation channel, in the HGN1 neurons and recorded the excitatory junction potentials (EJPs) in the gut muscles before and after activating ChR2 by light. The gut muscles received tonic excitatory inputs (*Figure 2A*). Due to the fillet recording methods we used to gain access of anus sphincter muscles, this firing pattern might differ from those in intact animals. Illumination of the larval VNC with blue light caused a dramatic increase of EJPs in the anus sphincter muscles in the larva with ChR2 expression in the HGN1 neurons but not in the control animals (*Figure 2B,C*), providing evidence for HGN1 innervation of sphincter muscles. The PDF neurons have been previously shown to promote visceral muscle contractions (*Talsma et al., 2012*). Activation of PDF neurons expressing ChR2 with blue light also triggered a dramatic increase of EJPs in the anus sphincter muscles (*Figure 2B,C*), indicating that PDF neurons also play a role in regulating anal sphincter contractions, although PDF neurons do not directly form synapse with these muscles. This light-induced activation was absent in UAS or Gal4 control larvae and dependent on retinal, which is the chromosphere for ChR2 channels (*Figure 2B,C*).

To test whether direct activation of PDF or HGN1 neurons could trigger gut muscle contraction, we first expressed ChR2 in the PDF neurons and use a Minos insertion line to label the hindgut. Because the blue light used to excite GFP in the hindgut can also activate ChR2, we could activate the neurons expressing ChR2 and monitor the gut movements at the same time. Both the hindgut and anal sphincter contracted strongly (*Figure 2D*) upon stimulation. In contrast, with ChR2 expression in the HGN1 neurons, the anal sphincter but not hindgut contracted upon blue light stimulation of HGN1 neurons (*Figure 2D* and *Video 2*). The contractions were absent in larvae with only UAS-ChR2 or larvae fed with regular food without retinal.

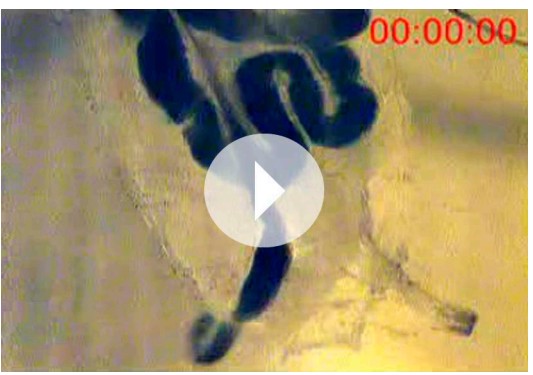

**Video 1**. Defecation behavior visualized with dyed feces. A larva was placed lateral side up on a slide. The animal was fed with dyed food and its feces can be seen when moving along the intestine.

## Functional requirement of PDF and HGN1 neurons in gut movements

To test whether PDF and HGN1 neurons are important for the normal defecation behaviors,

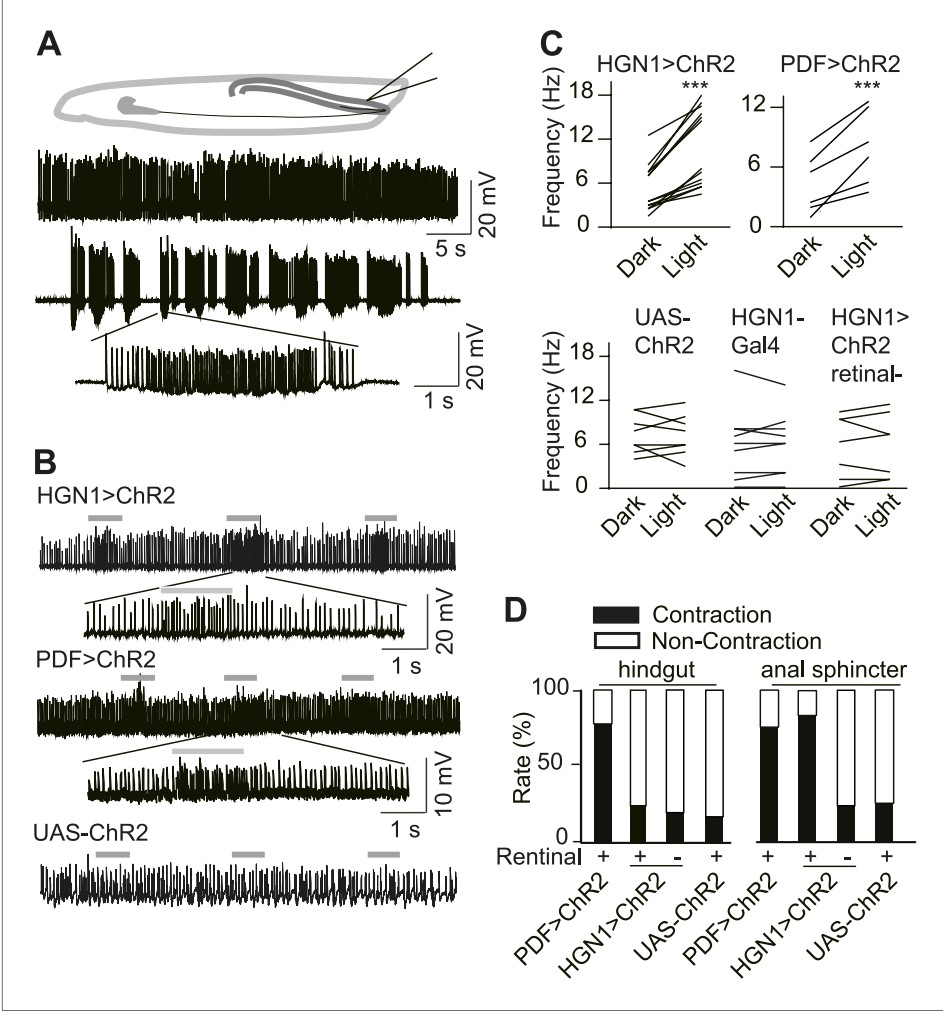

**Figure 2**. Excitatory output from VNC neurons to gut muscles. (**A**) Spontaneous EJP activity of the anus sphincter muscles. Top panel: tonic input to the muscles; middle panel: burst inputs recorded in the muscles; lower panel: zoom-in of the spikes in a single burst. (**B**) Light induced activation in the gut muscles. Top panel: light-triggered EJPs increase in the anus sphincter muscles of HGN1 > ChR2 larva. Grey bars indicate the blue light. Lower panel: light stimulation on UAS-ChR2 larva as control. (**C**) Paired plot of the EJP frequency in dark and light condition (n = 15, ***p < 0.001, paired *t* test). (**D**) Light-triggered gut contractions in larvae carrying ChR2 in their PDF or HGN1 neurons.

we expressed Kir2.1 in these neurons to inhibit their activities. Silencing PDF neurons caused the interval of the defecation to increase from 38 s to 94 s (*Figure 3A*). Silencing the HGN1 neurons did not significantly alter the interval of anus sphincter opening (*Figure 3A*). Conceivably the peristaltic hindgut movements driven by PDF neuronal activity could have generated sufficient pressure to force open the anus sphincter. Indeed, silencing both the PDF neurons and HGN1 neurons caused the larva to display barely any hindgut movement over 5 min thus rendering it difficult to estimate the defecation interval, in contrast to the nearly eight cycles of contraction over 5 min—corresponding to a defecation interval of 38 s—in control animals (*Figure 3B*). These results suggest that the PDF neurons and HGN1 neurons are required for the normal defecation behavior.

## Rhythmic activity of PDF and HGN1 neurons

To test whether the periodic defecation cycle of *Drosophila* larvae is associated with rhythmic firing of the PDF and HGN1 neurons innervating the hindgut and anal sphincter, we performed in vivo whole-mount Ca²⁺ imaging by monitoring the neuronal activity with a genetically encoded Ca²⁺ indicator GCaMP5 (*Tian et al., 2009*) driven by neuronal-specific Gal4s. Indeed, the PDF neurons displayed a

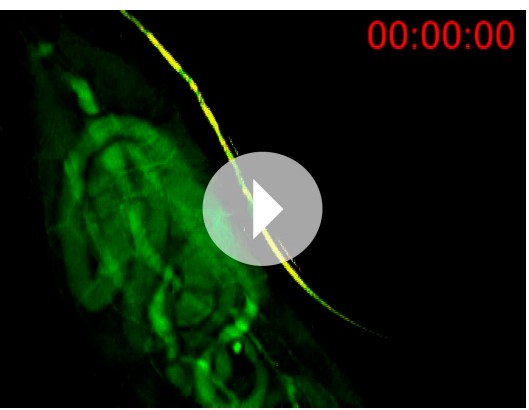

**Video 2**. Light-induced defecation via activation of ChR2 in HGN1 neuron. A larva was placed ventral side up on a slide. The defecation could be observed shortly after blue light illumination (as visualized by the auto-florescence of the internal organs).

stereotypical periodic firing pattern as revealed by $Ca^{2+}$ elevation in both soma and dendrite area (*Figure 4A,C*). The average interval between peak $Ca^{2+}$ signals is 38 s, which is highly consistent with the temporal pattern of contraction of the hindgut (*Figure 1D* and *Figure 4M*). We also monitored the $Ca^{2+}$ signals of the HGN1 neurons and found that the HGN1 neurons also exhibited oscillation of $Ca^{2+}$ levels with a periodicity of 38 s (*Figure 4B,D,M*, *Video 3*). Furthermore, $Ca^{2+}$ signals of HGN1 neurons spread from dendrites to soma with a fixed latency (*Video 3*), indicating that the HGN1 neurons may receive excitatory inputs with a periodicity of 38 s.

To further investigate the properties of HGN1 neurons, we performed $Ca^{2+}$ imaging and whole-cell patch clamp recording of these neurons in dissected VNC. HGN1 neurons exhibited periodic $Ca^{2+}$ activities though with less regularity in this isolated preparation (*Figure 4—figure supplement 1A,B*). They also displayed clusters of EPSCs (*Figure 4—figure supplement 1C*), indicating that they received excitatory input from upstream neurons. Whole-cell patch recording of HGN1 neurons revealed that they fired bursts of action potentials (*Figure 4—figure supplement 1D*), similar to what was seen in other *Drosophila* larval motor neurons (*Cattaert and Birman, 2001*; *Fox et al., 2006*; *Imlach et al., 2012*).

The arborizations of PDF neurons and HGN1 neurons overlap extensively along the midline of the posterior VNC (*Figure 4E,F* and *Video 4*), raising the question whether they interact with each other. Green fluorescent protein reconstitution across synaptic partners (GRASP) has been developed as a technique to indicate synaptic connection of two neurons, each expressing one component of the split GFP (*Feinberg et al., 2008*; *Gordon and Scott, 2009*; *Gong et al., 2010*; *Han et al., 2012*). We expressed the two GFP components separately in PDF neurons and HGN1 neurons by using two different binary expression systems, PDF-LexA and HGN1-Gal4. An intense GFP signal was observed

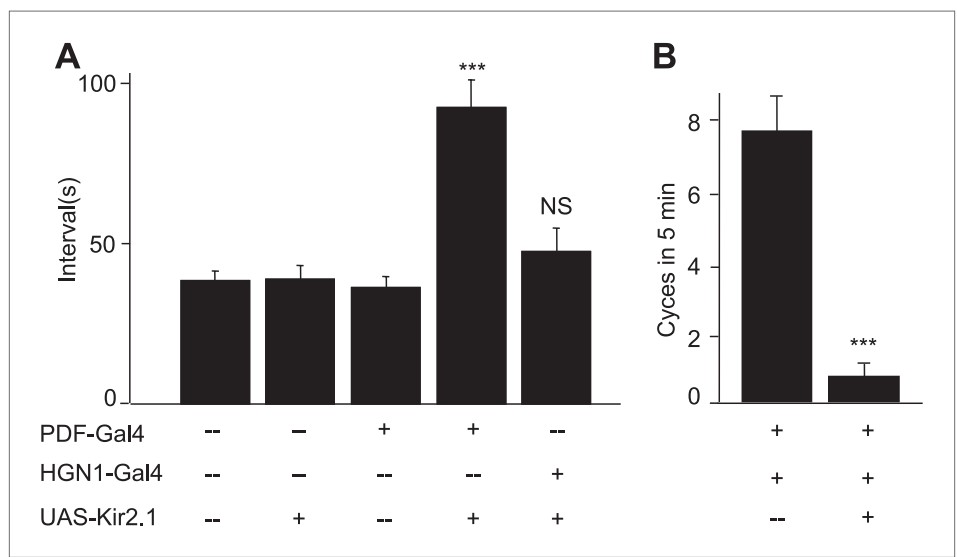

**Figure 3**. PDF and HGN1 motor neurons are essential for the defecation behavior. (**A**) Silencing PDF but not HGN1 neurons increased the defecation intervals (***p < 0.001, one-way ANOVA). (**B**) Silencing both PDF and HGN1 neurons reduced the defecation frequency dramatically (***p < 0.001, paired *t* test).

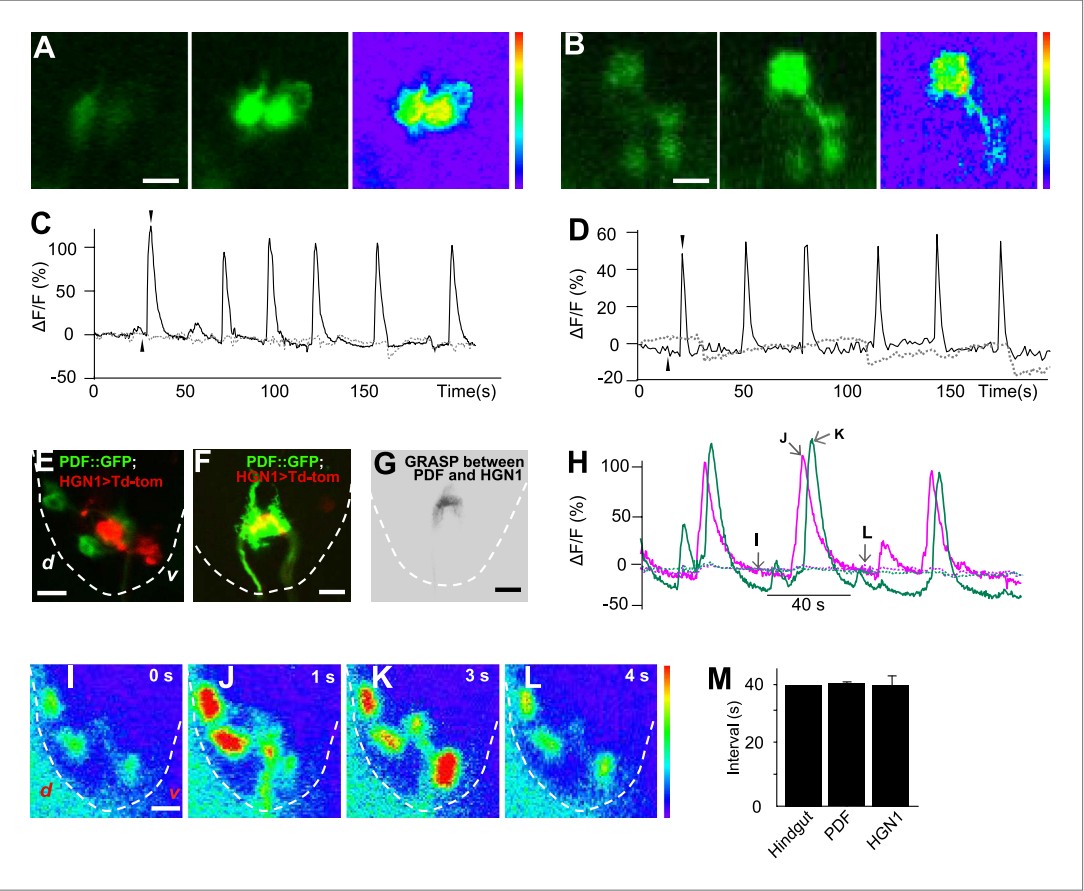

**Figure 4**. Sequential firing of the PDF and HGN1 neurons. (**A**) Spontaneous Ca²⁺ oscillation imaged by GCaMP5 in PDF neurons in vivo. From left to right, Ca²⁺ signal of PDF neurons in quiescent state; peak Ca²⁺ signal of PDF neurons; Ca²⁺ intensity increase (scale bar: 20 μm). (**C**) Ca²⁺ oscillates over time. Solid line: GCaMP5; dashed line: RFP, the two black arrow heads indicate the time points of left and middle panel in **A**. (**B**) Spontaneous Ca²⁺ oscillation imaged by GCaMP5 in HGN1 neurons in vivo. From left to right, Ca²⁺ signal of HGN1 neurons in quiescent state; peak Ca²⁺ signal of HGN1 neurons; Ca²⁺ intensity increase (scale bar: 20 μm). (**D**) Ca²⁺ oscillates over time. Solid line: GCaMP5; dashed line: RFP, the two black arrow heads indicate the time points of left and middle panel in (**B**). (**E**) PDF (green) and HGN1 (red) neurons at the lateral view of the VNC. d: dorsal; v: ventral. (**F**) Neuropil of PDF and HGN1 neurons overlap (scale bar: 20 μm). (**G**) Grasp signal that resembles the location and shape of neuropil co-localization (scale bar: 20 μm). (**H**) Spontaneous Ca²⁺ oscillation imaged by GCaMP5 in PDF (purple) and HGN1 (green) neurons simultaneously in vivo. Dashed line: RFP recorded at the same time. Arrows indicate the time points of the images as shown in (**I–L**). (**I–L**), representative images from **H**. Color bar shows the range (1–256) (scale bar: 20 μm). (**M**) Form left to right; intervals of defecation and Ca²⁺ oscillation of PDF and HGN1 neurons (error bar: S.E.M., n = 8, 7, and 7).

The following figure supplements are available for figure 4:

**Figure supplement 1**. HGN1 neurons' rhythmic activity in dissected VNC.

**Figure supplement 2**. Single component of split-GFP did not show fluorescence.

**Figure supplement 3**. PDF neurons have axonal projections inside the VNC.

in the area where the processes of these two groups of neurons overlap (*Figure 4G*), while neither PDF neuron nor HGN1 neuron expressing one part of the split GFP of GRASP generated any fluorescent signals by itself (*Figure 4—figure supplement 2*), suggesting that a functional connection might exist between PDF and HGN1 neurons. By labeling the PDF neurons simultaneously with the dendritic RFP marker DenMark and the axonal GFP marker sytGFP, we found that the PDF neurons send

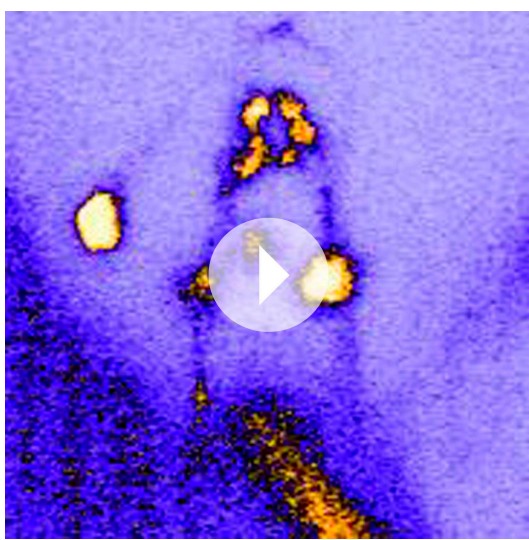

**Video 3**. Rhythmic activity of HGN1 motor neurons. A larva was placed ventral side up. GCaMP5 was driven by HGN1-Gal4 to monitor the neuronal activity. The video is at 6× speed.

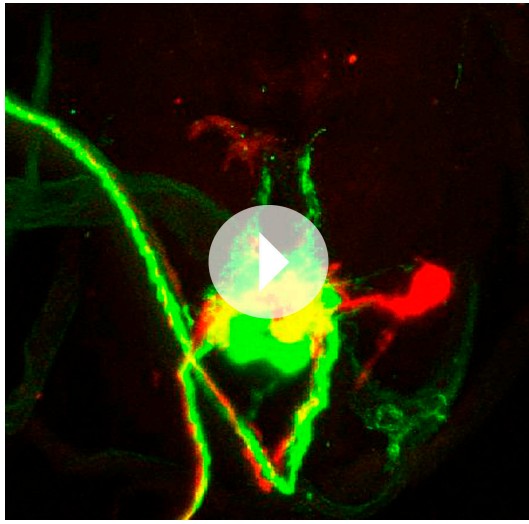

**Video 4**. 3-D reconstructions of PDF and HGN1 neurons in the VNC. PDF neurons are labelled with PDF-GFP. HGN1 neurons are labelled with HGN1-Gal4 driven tdTomato. The Z-stack images were taken with 2-μm optical slice and projected along the Y-axis to get the stereo visualization of the structures.

their axons to the area where their processes overlap with HGN1 neuron dendrites (*Figure 4—figure supplement 3*).

## Sequential firing of PDF and HGN1 neurons

To test whether the sequential contractions of the hindgut and the anal sphincter are associated with sequential firings of the PDF neurons and HGN1 neurons, we employed two Gal4 drivers to express GCaMP5 in both PDF neurons and HGN1 neurons at the same time. The cell bodies of PDF neurons are near the ventral surface of the VNC, while the cell bodies of HGN1 neurons are more dorsal and posterior (*Figure 4E*), making it possible to distinguish the two groups of cell bodies when monitored laterally (*Figure 4E*). By monitoring the $Ca^{2+}$ signals in PDF neurons and HGN1 neurons simultaneously, we found that the $Ca^{2+}$ level began to rise in PDF neuron cell bodies and then spread to the area occupied by arborizations of both groups of neurons, followed by $Ca^{2+}$ elevation in the cell bodies of HGN1 neurons with a very short delay, indicating that these two groups of neurons have coordinated firing patterns (*Figure 4H–L* and *Video 5*).

## A sensory neuron as the sensor of anus movement

Next, we asked whether there are sensory neurons for sensing movements of the gut or anus. The dendritic arborization (da) neurons are primary sensory neurons, which cover the entire body wall of a *Drosophila* larva. They are important for sensing chemical, thermal, light, and mechanical stimulations (*Tracey et al., 2003*; *Xiang et al., 2010*; *Kim et al., 2012*; *Yan et al., 2013*). In the vicinity of the anal slit, we found a specialized PPK-Gal4-labeled neuron. The cell body of this neuron resides on the anterior side of the anal slit, and its dendritic arbors surround the entire anal slit (*Figure 5A*). The majority of its dendrites forms a thin layer of arbors and covers the body wall around the anal slit (*Figure 5A*). There are also some arborizations extending along the anus sphincter (*Figure 5A*). The axon of this sensory neuron joins the nerve bundle that includes the axons of other da neurons and projects to the terminal segment of the VNC.

This single anus sensory neuron (ASN) has its cell body stochastically located on one side of the midline but its dendrites are highly symmetric. When the anal slit opens, the dendrites are dramatically stretched (*Figure 5B*). To investigate whether this stretch could activate the ASN, we carried out in vivo imaging of the ASN $Ca^{2+}$ level with GCaMP5. We found that opening of the anal slit is accompanied with $Ca^{2+}$ elevation in both dendrites and soma of this neuron (*Figure 5C,D* and *Video 6*). The periodicity of this $Ca^{2+}$ response is similar to that of the movements of the anus and the oscillations of PDF neurons and HGN1 neurons in the VNC

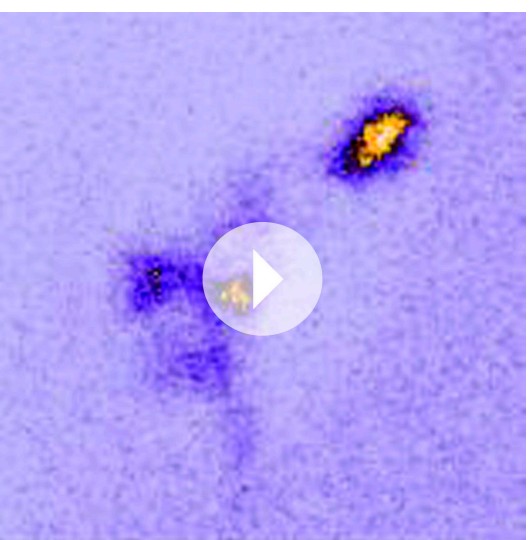

**Video 5**. Dual imaging of PDF and HGN1 neurons. The larva was placed lateral side up to visualize both groups of neurons. The neurons expressed GCaMP5 under the drive of PDF and HGN1 Gal4s. Left side: ventral; right side: dorsal. A $Ca^{2+}$ signal propagates from PDF neurons to HGN1 neurons.

(**Figure 5E**). This result indicates that the ASN is mechanosensitive and participates in sensing the radial stretch caused by opening the anus.

We next tested whether the activation of the ASN could affect the firing patterns of motor neurons in the VNC. This sensory neuron projects to the terminal segments of the VNC and its axon terminals overlap with the area occupied by the dendrites of the HGN1 neurons and PDF neurons (**Figure 5F**). To investigate whether there might be direct synaptic contact between these neurons, we carried out GRASP analysis by expressing components of the split GFP in PDF neurons and the ASN. We found that there is very strong GRASP signal at the site where ASN axons and PDF dendrites overlap (**Figure 5G**), while neither PDF neurons nor ASN expressing one part of the split GFP of GRASP generated any fluorescent signals (**Figure 5—figure supplement 1**). Interestingly, the GRASP signal is asymmetric, in concordance with the localization of the ASN cell body to one side of the midline leading to a more intense axonal projection to the ipsilateral VNC, which is also evident with asymmetric $Ca^{2+}$ elevation of the ASN axon terminals in the VNC (**Video 7**).

To further determine whether there is feedback from the ASN to activate the motor neurons in the VNC, we employed PPK-Gal4 to drive expression of ChR2 in the ASN so as to activate this neuron by blue light and monitored the EJPs of the gut muscles innervated by these motor neurons. Activation of ASN via blue light illumination induced large increases of EJPs in the majority of the anus sphincter muscles (**Figure 5H,I**).

To confirm that ASN rather than other PPK-Gal4-labelled neurons provides the feedback, we imaged PDF neurons and HGN1 neurons with GCaMP5 while inserting a tapered glass probe and advancing it to split open the anus sphincter, in a manipulation that mimicked the anus opening during defecation. We found that both PDF and HGN1 neurons responded to this local stimulation (**Figure 5J**). Stimulation of the anus with this glass probe induced an asymmetric $Ca^{2+}$ increase in the VNC that is consistent with the asymmetric projection of ASN axons (**Figure 5—figure supplement 2**).

To study the functional importance of the ASN feedback to motor neurons, we used 2-photon laser to ablate the cell body of ASN at 48 hr after egg laying (AEL). The ASN was completely abolished 48 hr after laser ablation, while the other PPK neurons remained intact (**Figure 5K**). We then imaged the cell bodies of HGN1 neurons in the VNC and found abnormality of their rhythmic firing pattern, which displayed a much longer interval compared to control animals (**Figure 5L**).

These results suggest that the central neurons in the VNC receive excitatory feedback from ASN to increase motor neuron firing. Together with the GRASP analysis revealing the physical proximity of the processes of ASN and PDF neurons, our results indicate that the sensory neuron in the anus might sense the stretch when the anal slit is open and respond by provide feedback modulation of the PDF neurons and HGN1 neurons in the central nervous system.

## NOMPC is required for mechanotransduction in the ASN

To search for the putative mechanotransduction channel in the ASN, we examined the defecation behavior of several mechanotransduction channel mutants. We found that the defecation rhythm remains normal in the *iav*, *nan*, and *piezo* mutants. However, the defecation interval in the *nompC* mutants was significantly increased (**Figure 6A**). NOMPC is a TRPN channel essential for adult hearing and larval gentle touch (**Gong et al., 2004**; **Yan et al., 2013**). We found that NOMPC is highly expressed all over the dendrites and soma of the ASN, as revealed by staining with antibody against the NOMPC protein (**Figure 6—figure supplement 1**), raising the possibility that it might play a role

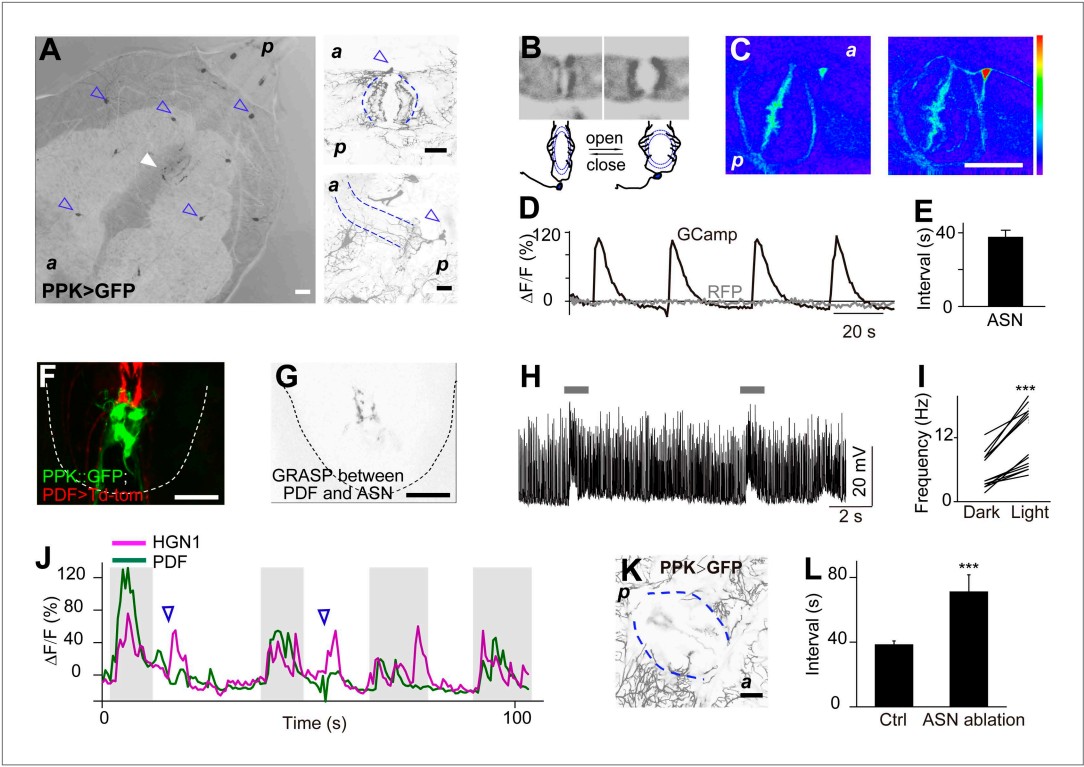

**Figure 5**. Sensory feedback from anus sensory neuron to the VNC motor neurons. (**A**) Anus sensory neuron's (ASN) location and morphology. Left panel: filled arrowhead indicates the location of the ASN; open arrowheads indicate other PPK-Gal4 neurons. Upper right panel: arrowhead: cell body; lower right panel: lateral view of the dendritic extension along the anus sphincter. Dashed line indicating the border of the intestine (scale bar: 50 μm). a: anterior; p: posterior. (**B**) Images and schematic drawing of the anus opening, showing the stretching of the ASN dendrites. (**C**) Ca²⁺ elevates in ASN when anus opens. Left panel: resting state when the anus is closed; right panel: anus is open and the dendrites are stretched. (Color range: 1–256. Scale bar: 50 μm). (**D**) Plot of the Ca²⁺ intensity of the cell body region. Black trace: GCaMP5 signal; grey line: tomato signal as control in the same neuron. (**E**) Peak-to-peak interval of ASN Ca²⁺ activity (error bar: S.E.M., n = 7). (**F**) Dual labeling of PPK and PDF neurons. Red: PPK neuron axon terminals (PPK–GFP); green: PDF neuron cell bodies and dendrites (PDF-Gal4 > td-Tomato) (scale bar: 20 μm). (**G**) GRASP signals between ANS/PDF neurons (scale bar: 20 μm). (**H**) Increase of EJPs in the anus sphincter muscles when the ASN carrying ChR2 is activated by blue light (indicated by grey bars). (**I**) Paired plots of the EJPs frequency in dark and light condition (n = 14, ***p < 0.001, paired t test). (**J**) Direct ASN stimulation triggered PDF/HGN1 neurons' activity (arrow head); grey bars indicate spontaneous oscillations of PDF/HGN1 neurons. (**K**) Laser ablation of ASN eliminated the single neuron (ASN) while other PPK neurons remained intact. Larval genotype: HGN1-Gal4; UAS-GCaMP5/PPK-tdTomato. (**L**) ASN ablation increased HGN1 neurons' activity interval.

The following figure supplements are available for figure 5:

**Figure supplement 1**. Single component of split-GFP did not show fluorescence.

**Figure supplement 2**. Direct ASN stimulation triggered asymmetric activation of PPK neurons' axon terminals.

in ASN sensing of radial stretch. Indeed, we found that the Ca²⁺ response is absent in the ASN of *nompC* null mutant larvae (***Figure 6B,C***, ***Video 8***). This defect could be rescued by expressing wild-type NOMPC in the ASN with PPK-Gal4 (***Figure 6B,C***, ***Video 9***). Our study thus identified a new role of the NOMPC channel, namely for sensing redial stretch of the intestinal terminus. NANCHUNG (NAN) and INACTIVE (IAV), two other TRP channels that often work in concert with NOMPC in other sensory neurons, are thought to form heterodimers and function in the mechanotransduction in *Drosophila* (***Gong et al., 2004***). We also tested the role of IAV in the ASN's stretch sensing. We found the ASN of *iav¹* mutant larvae exhibited Ca²⁺ response comparable to that in the wild-type larvae (***Figure 6C*** and ***Video 10***), suggesting that IAV is not required for the mechanotransduction of ASN.

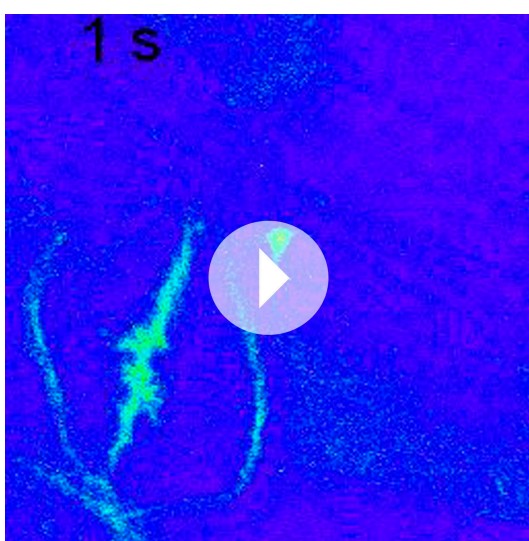

**Video 6**. ASN's response to anus opening in a wild-type larva. The ASN was labelled with PPK-Gal4 driven GCaMP5. A dramatic increase of $Ca^{2+}$ over dendrites and soma was observed when the anus opened.

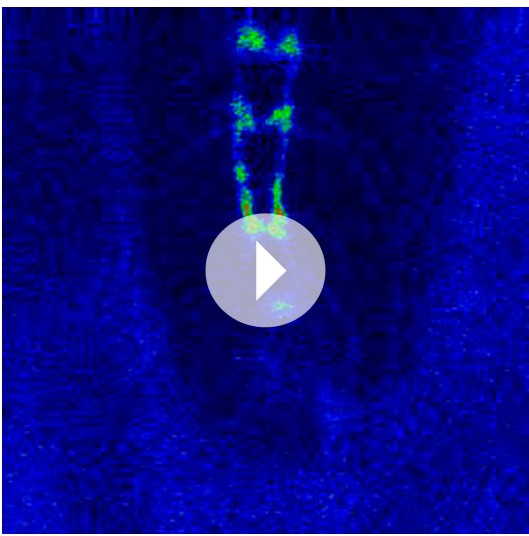

**Video 7**. Asymmetric activity of ASN axon projection in the VNC. The sensory neuron axons were labelled with PPK-Gal4 driven GCaMP5. Note the last segment exhibited intense activity increase and the signal on the right side was stronger that the left side, resulted from the asymmetric projection of the ASN.

## Discussion

### A new model system to study the defecation behavior

This study establishes the *Drosophila* larva as a model system for studying the defecation behavior. We found that *Drosophila* larvae exhibit periodic defecation cycles, involving sequential contractions of the hindgut and the anal sphincter. We also found two groups of neurons, which innervate the hindgut and anal sphincter respectively, and can excite the hindgut and anal sphincter muscle in a sequential manner. In addition, we found a single sensory neuron that could sense the opening of the anal slit and send feedback to the motor neurons (*Figure 7*). Studies of *C. elegans* as a model system have investigated the defecation circuit (*Thomas, 1990*; *Avery and Thomas, 1997*; *Branicky and Hekimi, 2006*; *Kwan et al., 2008*). Studies of the adult fly have identified neurons regulating defecation behaviors subject to dietary and reproductive modulation (*Cognigni et al., 2011*). In this study of the defecation behavior in *Drosophila* larvae, we have identified not only the motor neurons innervating gut muscles but also a sensory neuron strategically located to sense radial stretch during defecation and provide feedback to the central nervous system.

### Different features of larval and adult defecation behaviors

Previous studies of the defecation behaviors of the adult fly (*Cognigni et al., 2011*) have revealed that its defecation rate is regulated by both the internal state and environment, rather than showing a robust rhythm. However, at the larval stage, the motor neurons and gut muscles as well as the sensory neuron responding to anus movement, all show very robust rhythmic activities. Given that feeding and defecation are dominant behaviors for third-instar larvae, conceivably robust rhythmic feeding and defecation behaviors may facilitate their nutrition intake and waste expulsion. In contrast, adult flies will likely encounter more complex environments and may need to conduct their defecation behaviors in a more controllable manner.

### Mechanosensation involved in the defecation behavior

Mechanosensation serves a number of important physiological functions in *Drosophila* larvae. The radial stretch sensation is a special type of mechanosensation essential for the function of many organs with luminal structures such as the digestive system and the blood vessels. However, how the organs sense radial stretch remains unclear.

We have identified a sensory neuron that can sense radial stretch with its highly specialized morphology in *Drosophila* larvae. In addition, we found that the TRP channel NOMPC but not other TRP

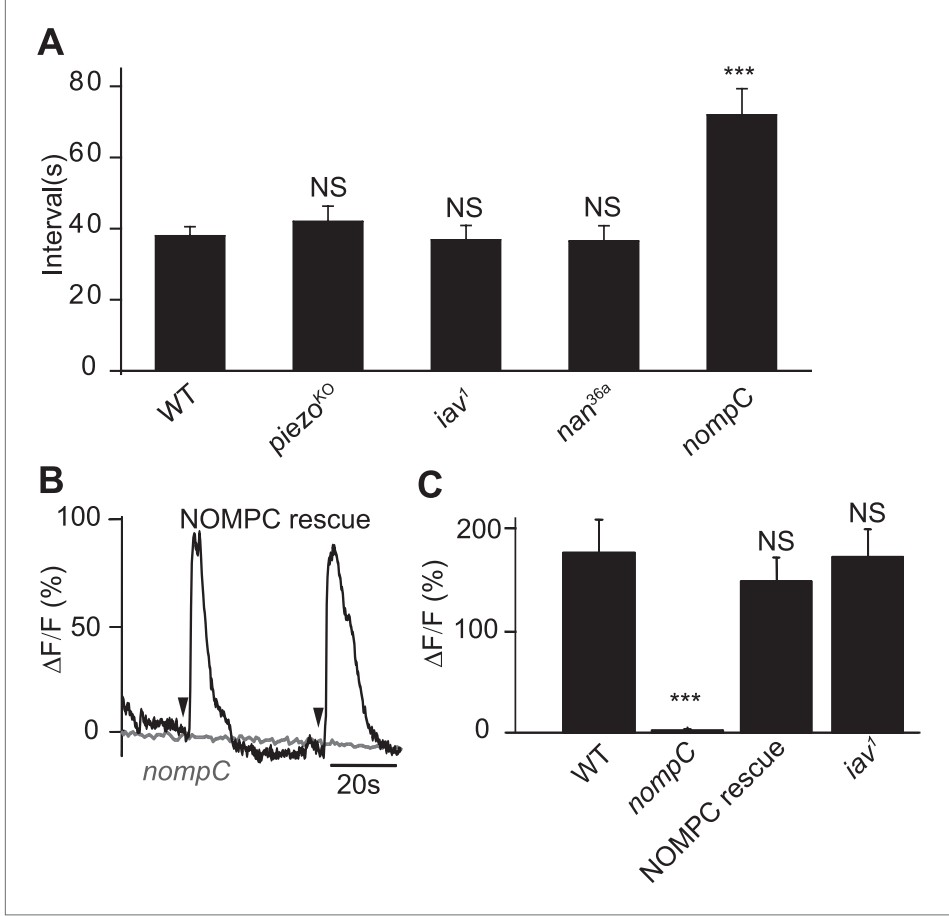

**Figure 6**. NOMPC is required for the mechanotransduction of ASN. (**A**) *nompC* but not *iav*, *nan*, and *piezo* mutant affects the defecation cycle of the larvae (error bar: S.E.M., n = 6, 7, 7, and 6. ***p < 0.001, one-way ANOVA). (**B**) $Ca^{2+}$ activity in the ASN responding to anus opening in *nompC* null mutant and NOMPC rescue larvae. Arrowheads indicate the anus opening for the rescue larva. (**C**) Group data of the $Ca^{2+}$ response of ASN (error bar: S.E.M., n = 7, 10, 9, and 9. ***p < 0.001, one-way ANOVA).

The following figure supplement is available for figure 6:

**Figure supplement 1**. NOMPC expression over the ASN revealed by NOMPC staining.

channels tested, such as IAV that is often associated with NOMPC function, is required for normal ASN mechanotransduction. Interestingly, the ASN could be labeled by both class III da neuronal marker and class IV da neuronal maker, raising the question whether it might have the dual functions to sense different stimuli. The ASN may provide a neuronal model to study the distinct and cooperative roles of different channels in a single neuron in the sensing of different intensity of stimulation.

## An entry point to study *Drosophila* larval defecation circuitry

The two motor neurons and the sensory neuron ASN provide an entry point to elucidate defecation circuitry. The two motor neurons appear to be functionally connected, possibly involving synaptic connections between them, although we cannot exclude the possibility of multiple neurons being engaged in their functional connections. It remains to be determined as to how they are entrained with this rhythmic firing pattern, and whether it involves a central pattern generator upstream of PDF neurons. Interestingly, PDF is a peptide that has important roles in multiple neuropeptide signaling pathways in the fruit fly (**Renn et al., 1999**; **Kim et al., 2013**); it would be interesting to test whether this neuropeptide also plays a role in the regulation of defecation behaviors by PDF neurons in the VNC. It is also of interest to explore possible contributions of indirect effects of PDF over muscle contraction, such as

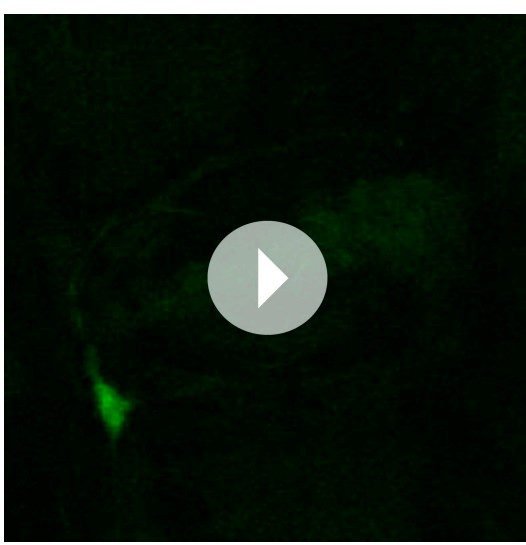

**Video 8**. *nompC* mutant eliminated ASN's response to anus opening. The ASN in the *nompC* mutant stayed quiescent even when it received a similar intensity of stretch. The genotype of the larva is: *nompC¹/nompC³*; NOMPC-Gal4/UAS-GCaMP5, UAS-mCherry.

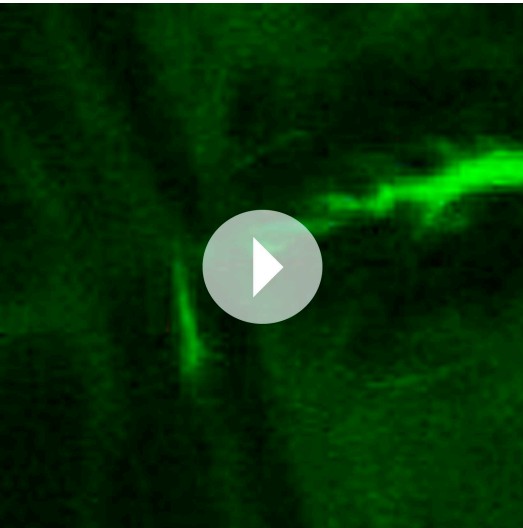

**Video 9**. NOMPC channel rescued the *nompC* mutant phenotype in ASN. A dramatic Ca²⁺ response could be detected when the wild-type NOMPC was expressed in the ASN with a *nompC* null background. The genotype of the larva is: *nompC¹/nompC³*; NOMPC-Gal4, UAS-NOMPC/UAS-GCaMP5, UAS-mCherry.

an influence of tracheal branching in the hindgut that may affect muscle contractions (*Linneweber et al., 2014*). Recently, a study has suggested a novel role of HGN1 neurons in regulating the long-term food intake behaviors of adult flies (*Olds and Xu, 2014*). In our study we found that HGN1 neurons control the rhythmic pattern of larval defecation. These two studies suggest that *Drosophila* HGN1 neurons at different developmental stages might have multiple functions in regulating feeding and defecation behaviors.

## Potential evolutionary conservation of defecation circuitry among different organisms

Though separated in evolution millions years ago, the structures of *Drosophila* gut and human gut share striking similarity. There are circular and longitudinal muscles lining the gut ending with the anal sphincter that controls defecation (*Netter, 1997*; *Murakami and Shiotsuki, 2001*). It remains an open question as to the extent of similarity of the mechanisms that control the gut movements. Diseases such as Hirschsprung's disease and anorectal malformation with failure to pass meconium (*Loening-Baucke and Kimura, 1999*) are caused by developmental abnormality related to the gut and its innervation. Several genes and specific regions on the chromosomes have been shown or suggested to be associated with Hirschsprung's disease. Mutations in two human genes could lead to the absence of certain nerve cells in the colon (*Puri and Shinkai, 2004*). With the powerful genetic tools, further study of the *Drosophila* larval gut rhythmicity and its neural modulation will help us identify evolutionarily conserved features as well as strategies that may have been adopted by different organisms for their fitness.

## Materials and methods

### Fly stocks

All the larvae were raised in the normal fly medium (for the light activation assay, 100 µM all-trans retinal was added to the food). Flies are kept in 12 hr/12 hr dark/light circle at 25°C. PDF-Gal4, HGN1-Gal4, and UAS-ChR2 are from Bloomington stock center. GRASP was done using lines: PDF-loxA > loxAop-mCherry and HGN1-Gal4 > UAS-GFP or PPK-Gal4 > UAS-GFP.

w[1118]; Gr28b[MB03888] is a Minos insertion which is from stock center (#24190), UAS-GCaMP5 fly line is from Loren L Looger lab in Janelia Farm. *piezo^KO* is from A Patapoutian lab in Scripps.

GRASP between PDF neurons and HGN1 neurons: w; PDF-Lexa/UAS-CD4-sp1-10; HGN1-Gal4/LexAOp-CD4-sp11. GRASP between PPK neurons and HGN1 neurons: w; +/UAS-CD4-sp1-10; HGN1-Gal4/PPK-tdTomato-sp11. UAS-CD4-sp1-10; LexAOp-CD4-sp11 are from K Scott lab (UC Berkeley).

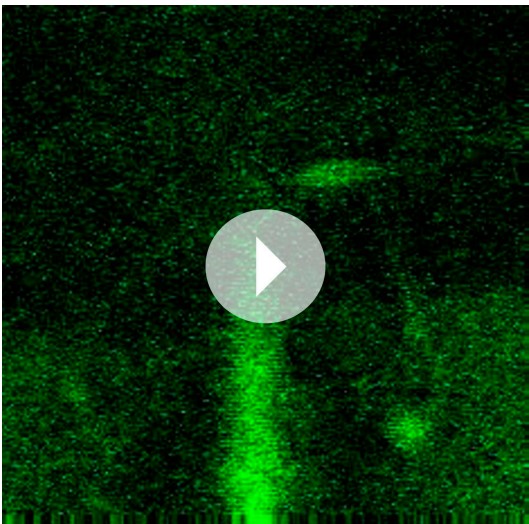

**Video 10**. IAV is not required for the ASN's stretch sensing. The ASN of *iav*[1] mutant larva exhibited similar response pattern to anus opening compared with wild-type larvae. The genotype of the larva is: *iav*[1]/y; NOMPC-Gal4/UAS-GCaMP5, UAS-mCherry.

## Whole-mounting imaging

For the whole-mounting imaging, a freely moving larva was picked up and rinsed with distilled water. Then the larva was transferred into 4% PFA overnight at 4°C. The larva was put between cover glass and images were taken by Zeiss confocal microscopy. In some cases, the whole VNC or different part of the gut was dissected out and mounted on a cover glass in PBS for imaging.

For immunostaining of *Drosophila* larvae, third instar larvae were dissected in PBS. The whole hindgut and anus were isolated from their bodies. The tissues were then fixed in 4% PFA solution for 20 min at room temperature and treated with the primary antibody (NOMPC antibody from J Howard [Yale], vGlut antibody from G Davis [UCSF]) overnight at 4°C and secondary antibody for 2 hr at room temperature. Images were acquired with Leica SP5 confocal microscope.

## Behavioral assay

### Defecation assay

Third instar larvae were picked up, rinsed, and transferred on to agar plate with yeast paste supplied with food dye (FD&C blue 1 and red 40, 1:1000). The larvae were fed with food dye for 2 hr and mounted between cover glasses for experiment.

### Gut movement assay

A hindgut-specific byn-Gal4 was crossed with UAS-GFP to visualize the hindgut. A third instar larva was gently picked up from the food surface and rinsed with distilled water briefly. The larva was then transferred into a drop of PBS on the slide. A cover glass was put on the larva and pressed slightly to reduce larval movement. The larva was mounted ventral side up under a Leica stereoscope and video-taped for later analysis.

### Light activation assay

Mercury light filtered with a GFP filter was applied to the larva preparation with a certain duration of time. The movements of the hindgut and anus sphincter were video-taped for further analysis. The delay of contraction was calculated between the light onset and the anus sphincter contraction. Both hindgut and anus sphincter movements could be easily visualized with auto-fluorescence of food debris in the intestines.

## EJP recording

Free moving third instar larva was pinned onto Sylgard coated chamber dorsal side up and filleted along the dorsal body wall. The larva was dissected in a saline containing: (in mM): 103 NaCl, 3 KCl, 5 TES, 10 trehalose, 10 glucose, 7 sucrose, 26 $NaHCO_3$, 1 $NaH_2PO_4$, and 4 $MgCl_2$, adjusted to pH 7.25 and 310 mOsm. 2 mM $Ca^{2+}$ was added to the saline fresh before use. Fat bodies were gently removed from the gut surface. Additional pins were used to immobilize the gut. The preparation was visualized by Zeiss axioscope microscopy with 40× water lens. Sharp electrode with resistance around 80 MΩ was filled with 3 M KCl. The electrode tip was approached to the gut surface under the control of the MP-285 manipulator (Sutter, USA). The signal was acquired by the Axon 200B amplifier and filter at 2 kHz. The electrode was moved forward until the voltage suddenly dropped to around −40 mV.

Blue light was generated by mercury lamp with multiple filters. For ChR2 activation, a GFP filter was used to give out blue light with wavelength around 488 nm. The light application was controlled by a shutter equipped on the microscope. For pulse light activation, 2 s light pulse was repeated for three times. Recording data were analyzed with Clampfit and Matlab. The frequency before and during light application were calculated and compared as the index of activation.

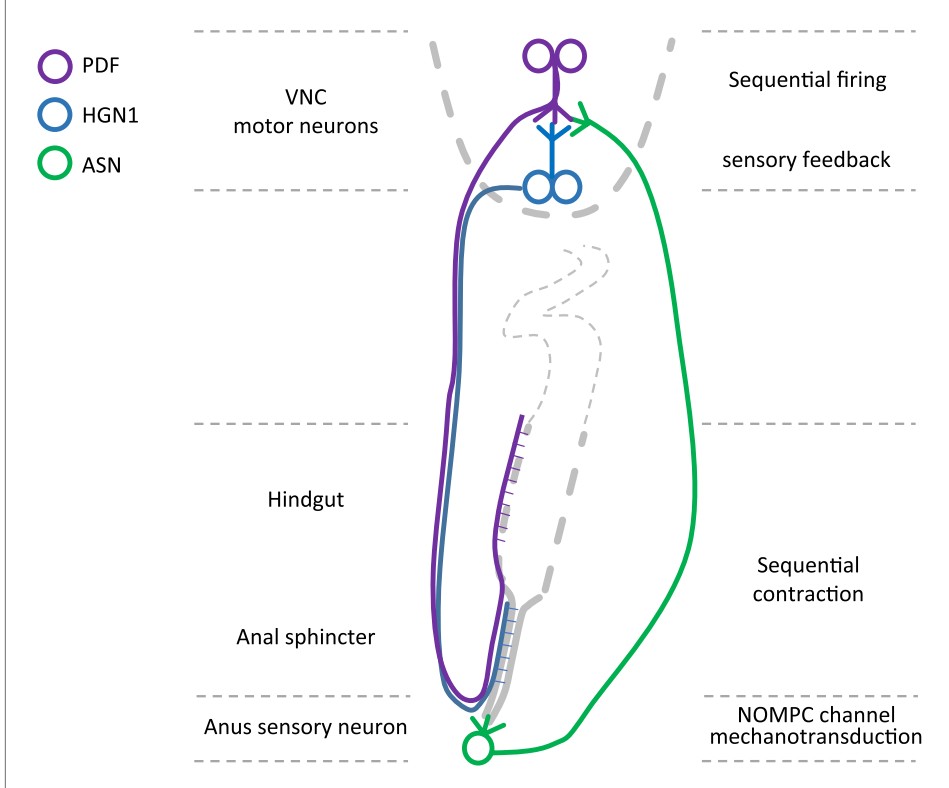

**Figure 7**. Graphic summary of the motor neurons and a mechanosensitive sensory neuron in the defecation circuitry.

## Whole-cell patch recording

The recordings were performed following the protocol described by *Hu et al. (2010)* with slight modifications. Briefly, the entire VNC of a third instar larva was dissected, and the peri-neural sheath was gently removed in recording saline containing 103 mM NaCl, 3 mM KCl, 5 mM TES, 10 mM trehalose, 10 mM glucose, 7 mM sucrose, 26 mM $NaHCO_3$, 1 mM $NaH_2PO_4$, 1.5 mM $CaCl_2$, and 4 mM $MgCl_2$ (adjusted to 280 mOsm, pH 7.3). The dissected VNC were transferred to a glass-bottom recording chamber containing recording saline and immobilized with a platinum frame. The HGN1 neurons were identified by their GFP signals under a 40× water objective. Current-clamp and voltage-clamp recordings were performed using patch-clamp electrodes (9–10 MΩ) filled with internal solution (140 mM potassium D-gluconate, 10 mM HEPES, 4 mM MgATP, 0.5 mM $Na_3GTP$, 1 mM EGTA, adjusted to 265 mOsm, pH 7.3). Cells were used for recording if the $R_m$ value was greater than 500 MΩ and the membrane potential value was lower than −50 mV. A small constant hyperpolarizing current was injected during recording, immediately after break-in, to bring the membrane potential of neurons to approximately −60 mV. Cells were held at −60 mV in voltage-clamp mode for EPSC recordings. Signals were acquired with an Axon-700B multiclamp amplifier and were digitized at 10 kHz and filtered at 2 kHz using a 1322A D-A converter. Data were analyzed using Clampex 9.0 software (Molecular Devices).

## ASN stimulation

A larva was mounted between two cover glasses with ventral side up. Its tail end was exposed for access of probe stimulation. A glass pipette was pulled and polished to form a taper-shaped probe with a diameter around 20 µm. The probe was spread on with grease to reduce friction. A piezo-controller was used to control the movement of the probe with fixed angle and increment. For imaging of ASN axons and PDF/HGN1 cell bodies, the probe was advanced 10 µm and the stimulation lasted for 1 s.

## ASN ablation

ASN ablation was carried out as previously described (*Song et al., 2012*). Briefly, a single second instar larvae 48 hr after egg laying (AEL) was mounted dorsal side up, and the cell body of the ASN was

targeted using a focused 930-nm two-photon laser (~350–700 mW) mounted on a custom-built Zeiss fluorescence microscope. Following lesion, animals were recovered on grape juice agar plates and imaged live at the appropriate stages.

## Ca²⁺ imaging

A third instar larva was gently picked up from the food surface and rinsed with distilled water several times. The larva was transferred into a drop of PBS on the slide. A cover glass was put on the larva and pressed slightly to reduce larval movement. The preparation was then put under the Zeiss Pascal 510 confocal microscopy equipped with a 20× air objective. Time series images were acquired and used for analysis.

For HGN1/PDF dual imaging, the larva was mounted with the lateral side up, so that both groups of neurons could be visualized simultaneously.

For PPK neurons imaging, larvae were transferred to a piece of filter paper saturated with 100 mM sucrose for 4–6 hr to remove the food debris in the gut which could have potentially covered the neuron's images during experiments.

An automatic alignment was made to most image series since the larvae tend to move slightly during image acquisition. An ImageJ plugin 'registration ROI' was utilized to correct the movements of the images during recording.

## Acknowledgements

We thank A Patapoutian and L Looger for fly lines. We thank J Howard and T Wang for antibodies. We thank S Younger, Y Song, M Klassen, and S Barbel for technical support. We thank members of the Jan lab for discussion. ZY was a recipient of the Long-Term Fellowship from the Human Frontier Science Program. ZY is supported by the Program for Professor of Special Appointment (Eastern Scholar) at Shanghai Institutions of Higher Learning and the Shanghai Rising-Star Program. This work was supported by NIH grants (R37NS040929 and 5R01MH084234) to YNJ. LYJ and YNJ are investigators of the Howard Hughes Medical Institute.

## Additional information

### Funding

| Funder | Grant reference number | Author |
|---|---|---|
| Howard Hughes Medical Institute | | Wei Zhang, Zhiqiang Yan, Lily Yeh Jan, Yuh Nung Jan |
| National Institutes of Health | NS069229 | Wei Zhang, Zhiqiang Yan, Lily Yeh Jan, Yuh Nung Jan |
| Human Frontier Science Program | | Zhiqiang Yan |

The funders had no role in study design, data collection and interpretation, or the decision to submit the work for publication.

### Author contributions

WZ, ZY, Conception and design, Acquisition of data, Analysis and interpretation of data, Drafting or revising the article; BL, Acquisition of data; LYJ, YNJ, Conception and design, Drafting or revising the article

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
