## [Decision Letter]

Thank you for sending your work entitled “Identification of motor neurons and a mechanosensitive sensory neuron in the defecation circuitry of Drosophila larvae” for consideration at *eLife*. Your article has been evaluated by Vijay Raghavan (Senior editor) and 2 reviewers, one of whom, Howard Baylis, has agreed to reveal his identity.

The Senior editor and the two reviewers discussed their comments before we reached the decision that the manuscript needs revisions, and the Senior editor has assembled the following comments to help you prepare a revised submission.

This is an interesting paper that seeks to define the defecation programme in *Drosophila* larvae and to identify neuronal and molecular substrates for this behaviour. The paper has three main topics in the results; characterisation of the defecation motor programme, characterisation of three neuronal populations which control hind-gut muscle contractions and identification and characterisation of a neuronal TRP channel involved in mechanosensory feedback. Some of the data is of high quality (e.g. calcium imaging experiments) and the potential finding of a mechanosensitive neuron involved in the process is novel and potentially interesting. The work could have wider implications for understanding defecation and the generation of rhythmic behaviour, but there are substantial concerns with the paper as it stands.

Contrary to what the manuscript seems to imply, defecation behaviour has been previously studied in flies (see Edgecomb J Exp Biol 1994 and Cognigni et al Cell Metab 2011). In particular, Cognigni et al developed a method that allows quantification of diuresis, enteric function, and food intake, and showed complex dietary and reproductive modulation of some of these features. The characterisation of the neuronal circuitry underlying defecation per se is useful but linking to the above features is of broader interest and should be attempted. Furthermore, contrary to what the authors state, only one of the three neuronal populations that they “identify” is novel in this context, given that: 1) The PDF neurons have been previously shown not only to innervate the hindgut (Nassel et al J Comp Neurol 1993, [43] PNAS), and also to promote visceral muscle contractions (Talsma et al PNAS 2012), and 2) The RN2 neurons have also been shown to innervate the hindgut and to regulate faecal output (Cognigni et al Cell Metab 2011, line referred to as HGN1-Gal4). [The authors should refer to the HGN1-Gal4 as this will allow other workers to connect the two studies.]

Both the Abstract and the body of the manuscript contain over-interpretations of the data and important omissions of previously published data. These need to be addressed in a revised manuscript. A revised manuscript should also show that what is being examined is indeed a circuit and that it controls actual defecation (as opposed to subtly modulating muscle contractions of the hind-gut). The experiments suggested below are achievable in a reasonable timeframe of about three months if the authors have the tools to manipulate those neurons with cellular resolution (all three drivers are much more broadly expressed so the observed behavioural effects, if any, could be secondary to interfering with all those other neurons). Substantial points are elaborated and listed below.

Characterization of “defecation behaviour” and its rhythmicity. The authors infer, but never show, effects of these neurons on defecation based on their effects on hindgut muscle contractions. Defecation rate could be subject to additional layers of neuronal control involving both visceral and somatic muscles, so the authors would need to show that these neurons do regulate defecation by, for example, using Fluoropoo (Cognigni et al Cell Metab 2011) if it works in larvae or by loading larval guts with dye-supplemented yeast and then recording defecation rate on unlabelled yeast. Related to this, the authors often make statements about defecation “rhythms/cycles” akin to those reported in C. elegans. These are based, as far as one can tell, on the rhythmic Ca2+ oscillations observed in resting PDF and RN2 neurons. However, the muscle recordings are not particularly rhythmic in resting conditions, and no evidence for rhythmic defecation events has been provided. For example, the pattern of EJPs shown in Figure 2 is described as rhythmic, however whilst it looks sporadic it does not look overtly rhythmic, analysis should be performed to support this statement and in support of a correlation between hind-gut contraction and defecation itself. Previously published data (Edgecomb J Exp Biol 1994 and Cognigni et al Cell Metab 2011) argues against a robust rhythm at least in adults, where defecation rate is regulated by both environmental (e.g. dietary content) and internal (e.g. reproductive) state. Thus, the conditions in which rhythmic defecation is observed should be precisely described.

Figure 3 which describes defecation data is not clear. What is being measured? The data in Figure 3 is described as hind-gut contraction, but 3A is simply described as “defecation” in the text. “Defecation” is too vague in this context, the exact step that is being measured needs to be defined. If the two measurements are the same thing, which one would hope, then the data in Figure 3 should be presented using the same measure, i.e. either both should be period in seconds or both should be defecation per period. This reflects a need as mentioned above to give precise details of the behavioural assays and data.

Gut peristalsis and defecation rate may vary depending on the larval stage. The authors seem to do all their experiments in L3 larva. Were these early, post-feeding, post critical weight or wandering L3s? Quantifications would need to be confined to one of these substages or data should be provided showing no differences between substages. Again there is a need for precise details of the behavioural assay and data.

Loss and gain-of-function phenotypes of PDF and RN2 neurons. Loss-of-function experiments were carried out constitutively throughout embryonic and larval development using Pdf-Gal4 and RN2-Gal4. Both drivers are expressed in other neurons that, when inactivated, could affect behaviours (e.g. feeding, circadian) that could indirectly impact gut movements (Thus, getting these experiments done at cellular resolution is important). Importantly, PDF has been shown to regulate tracheal branching in the hindgut, which could also affect muscle contractions (Linneweber et al Cell 2014). In the GoF muscle physiology experiments, no “raw” data has been provided for PDF neurons, and different experiments seem to have been conducted for the two neuronal populations. Only one control seems to have been used for the ChR2 experiments – several are normally required (+/- all-trans retinal, Gal4 and UAS parental controls). More details would also have to be provided about quantifications (how are the effects of tissue compression on fluorescence controlled? The hindgut diameter should become smaller. How would the ROI remain the same if the gut is actively contracting?).

The authors state that both PDF and RN2 neurons are motor neurons, based on comparisons with the gut innervation seen using VGLUT-Gal4. This is unconvincing. The authors should show the positions of the PDF, RN2 and OK371 axons on the hindgut muscles side by side, and use a broad neuronal marker like 22c10 to co-label the major nerves. What functional data shows a motor rather than neuromodulatory role for these neurons, especially for PDF?

The role of RN2 (see above too) is unclear. Activation of RN2 by ChR2 causes EJPs in the anal sphincter. This is the only direct evidence for the proposed role of RN2 as exciting the anal sphincter muscles. Other evidence suggests a more complex situation, ablation of RN2 does not prevent sphincter opening but does act additively with ablation of PDF on hindgut contraction period (Figure 3, also see comments re Figure 3 below). So does RN2 activation, by ChR2, alone have an effect on hind gut peristalsis? This data should be added to Figure 2.

The sequential firing is very neat. There are several experimental predictions that come from this. One is that activation of PDF by ChR2 should induce anal muscle contraction (via RN2), data should therefore be included in Figure 2 on the anal sphincter muscle contraction when PDF is activated by ChR2. Similarly activation of PDF by ChR2 would be expected to give rise to anal sphincter EJPs. Another expectation is that inactivation of PDF may result in no Ca2+ signals in RN2, this could be tested by combining available reagents. Finally would it also be possible to test whether activation of PDF using ChR2 led to calcium signals in RN2?

The authors seem to have used HGN1-Gal4 to label the RN2 neurons. As described in Cognigni et al, HGN1-Gal4 is a particular line (the D insertion) resulting from mobilizing the initial RN2-Gal4 (Landgraf et al). This is an important distinction because other RN2-Gal4 lines are either expressed more broadly, or are not expressed in those hindgut-innervating neurons and HGN1-Gal4 should be used in the text as this will allow other workers to connect the two studies.

The data showing that PDF neurons are presynaptic to RN2 is not very convincing (especially the sytGFP/Denmark figure) - both neurons could be putting out dendrites in the same VNC region in order to respond to the same presynaptic partner.

The sensory role of the ASN is potentially interesting, but there is no data linking that particular ppk neuron (as opposed to other ppk+ neurons) with the hindgut muscle contraction phenotypes and/or defecation control. The authors could use the elegant approach previously used by this lab (Yang et al Neuron 2009, Figure 3) to strengthen this link.

ASN is identified as sensory neuron providing mechanosensory feedback from the anus. ASN shows periodic Ca2+ oscillations. These correlate to the opening of the anus and activity in PDF and RN2, however no functional test of the hypothesis that ASN activation lies downstream of PDF, RN2 and anal opening is shown. One experiment would be to show that inactivation of PDF results in a loss of Ca2+ signals in ASN. A more positive experiment would be to show that activation of RN2 (using ChR2 or dTRPA1) in turn activates ASN. The feedback experiments use a very indirect test of feedback onto PDF and/or RN2, EJPs in the anal sphincter muscles. This may be acting through ASN-PDF-RN2-anal sphincter muscle series. However there are other explanations, in particular it does not rule out other routes of ASN feedback onto the anal sphincter muscles. It would be much better to test the effect of ASN activity more directly on PDF. For example do NOMPC mutants have changes in PDF Ca2+ signals, which are alleviated by NOMPC rescue in ASN? Does ablation of ASN change PDF calcium signals? Even better would be activation of ASN and direct measurement of signals in PDF. This would add significant weight to the argument that ASN is a functional modify of PDF properties.

---

## [Author Response]

Major additions and changes made in the revised paper:

1**)** We have extensively cited the work on adult fly defecation (Cognigni et al., Cell Metab 2011) and clarified the correspondence of our RN2-Gal4 with HGN1-Gal4 by using the latter terminology in the revised manuscript. We have also compared our study of larval defecation with previous studies of defecation of the adult fly in Discussion.

2) We have conducted additional experiments to test whether PDF and HGN1 neurons display certain features characteristic of motor neurons. 1) These neurons express vGlut, which is a motor neuron marker in *Drosophila*; 2) PDF neurons could be double labeled with vGlut-Gal4 (shown in the new Figure 1). 3) HGN1 neurons exhibit burst firing, similar to what was seen for other *Drosophila* motor neurons.

3) We have performed additional experiments to show ASN is necessary to modulate motor neuron activities. We used laser ablation to specifically kill ASN without affecting other neurons and observed a reduction of HGN1 rhythmic activity (shown in the new Figure 5).

4) We have shown ASN activation is sufficient to induce activity of PDF and HGN1 neurons by mechanically stimulating this neuron with a probe without affecting adjacent neurons (shown in the new Figure 5).

5) We have included additional control experiments and detailed methods in the revised manuscript.

*This is an interesting paper that seeks to define the defecation programme in Drosophila larvae and to identify neuronal and molecular substrates for this behaviour. The paper has three main topics in the results; characterisation of the defecation motor programme, characterisation of three neuronal populations which control hind-gut muscle contractions and identification and characterisation of a neuronal TRP channel involved in mechanosensory feedback. Some of the data is of high quality (e.g. calcium imaging experiments) and the potential finding of a mechanosensitive neuron involved in the process is novel and potentially interesting. The work could have wider implications for understanding defecation and the generation of rhythmic behaviour, but there are substantial concerns with the paper as it stands*.

*Contrary to what the manuscript seems to imply, defecation behaviour has been previously studied in flies (see Edgecomb J Exp Biol 1994 and Cognigni et al Cell Metab 2011). In particular, Cognigni et al developed a method that allows quantification of diuresis, enteric function, and food intake, and showed complex dietary and reproductive modulation of some of these features. The characterisation of the neuronal circuitry underlying defecation per se is useful but linking to the above features is of broader interest and should be attempted. Furthermore, contrary to what the authors state, only one of the three neuronal populations that they “identify” is novel in this context, given that: 1) The PDF neurons have been previously shown not only to innervate the hindgut (Nassel et al J Comp Neurol 1993,*
[43]
*PNAS), and also to promote visceral muscle contractions (Talsma et al PNAS 2012), and 2) The RN2 neurons have also been shown to innervate the hindgut and to regulate faecal output (Cognigni et al Cell Metab 2011, line referred to as HGN1-Gal4). [The authors should refer to the HGN1-Gal4 as this will allow other workers to connect the two studies*.*]*

We agree that the Cognigni et al Cell Metab paper has pioneered the study of defecation behaviors of fly, and we should have included more extensive discussion and reference to this study. In this revised manuscript, we have cited this study extensively and attempted to make connection between our study and previous studies. We have also cited the other relevant papers as the reviewers have suggested and made modification in the text accordingly.

*Both the Abstract and the body of the manuscript contain over-interpretations of the data and important omissions of previously published data. These need to be addressed in a revised manuscript. A revised manuscript should also show that what is being examined is indeed a circuit and that it controls actual defecation (as opposed to subtly modulating muscle contractions of the hind-gut). The experiments suggested below are achievable in a reasonable timeframe of about three months if the authors have the tools to manipulate those neurons with cellular resolution (all three drivers are much more broadly expressed so the observed behavioural effects, if any, could be secondary to interfering with all those other neurons). Substantial points are elaborated and listed below*.

*Characterization of “defecation behaviour” and its rhythmicity. The authors infer, but never show, effects of these neurons on defecation based on their effects on hindgut muscle contractions. Defecation rate could be subject to additional layers of neuronal control involving both visceral and somatic muscles, so the authors would need to show that these neurons do regulate defecation by, for example, using Fluoropoo (Cognigni et al Cell Metab 2011) if it works in larvae or by loading larval guts with dye-supplemented yeast and then recording defecation rate on unlabelled yeast*.

The methods used in this paper (Cognigni et al Cell Metab 2011) are very elegant. We have tried to use a similar approach to study larval defecation as the referees suggested, but failed for the following two reasons: 1) the feces of fly larvae are very tiny, making it hard to quantify; 2) the larval feces are actually thick liquid rather than solid. They got smeared and wiped away when larvae crawled over them.

For an alternative approach, we took advantage of the transparency of the larval body wall and visualized defecation process by labeling the fly food with a blue food dye following the referees’ suggestion. The defecation cycle revealed by the expulsion of blue feces is similar to what we have seen in our previous experiments. These data are included in the new Figure 1 and Video 1.

*Related to this, the authors often make statements about defecation “rhythms/cycles” akin to those reported in C. elegans. These are based, as far as one can tell, on the rhythmic Ca2+ oscillations observed in resting PDF and RN2 neurons. However, the muscle recordings are not particularly rhythmic in resting conditions, and no evidence for rhythmic defecation events has been provided. For example, the pattern of EJPs shown in*
Figure 2
*is described as rhythmic, however whilst it looks sporadic it does not look overtly rhythmic, analysis should be performed to support this statement and in support of a correlation between hind-gut contraction and defecation itself*.

We agree with the referees that the gut muscle activity as revealed by EJP recordings should have rhythmicity similar to that of the firing of PDF and HGN1 neurons and defecation. However, our recording from dissected larvae may not capture accurately the features of motor activities in intact animals. The purpose of showing these data is to demonstrate that gut muscles receive excitatory input from motor neurons. We have clarified this point by rephrasing the descriptions of the EJP recordings and making note of the likely difference in firing patterns observed in fillet preparations compared to intact animals in the revised manuscript.

*Previously published data (Edgecomb J Exp Biol 1994 and Cognigni et al Cell Metab 2011) argues against a robust rhythm at least in adults, where defecation rate is regulated by both environmental (e.g. dietary content) and internal (e.g. reproductive) state. Thus, the conditions in which rhythmic defecation is observed should be precisely described*.

We appreciate the suggestion of the referees. We think the defecation behaviors in larvae could exhibit a more robust rhythm, which is different from that in adult flies, possibly because larvae spend the majority of their time consuming food to sustain their rapid growth whereas adult flies may need to command greater control of their feeding and defecation behaviors to cope with a more complex environment. It is very possible that larval defecation behavior is also modulated by other cues; however, exploration of such modulation is beyond the scope of this study which focuses on neuronal control of rhythmic defecation cycles. We have also discussed this point and compared the larval and adult fly behaviors in the revised text.

Figure 3
*which describes defecation data is not clear. What is being measured? The data in*
Figure 3
*is described as hind-gut contraction, but 3A is simply described as “defecation” in the text. “Defecation” is too vague in this context, the exact step that is being measured needs to be defined. If the two measurements are the same thing, which one would hope, then the data in*
Figure 3
*should be presented using the same measure, i.e. either both should be period in seconds or both should be defecation per period. This reflects a need as mentioned above to give precise details of the behavioural assays and data*.

We thank the referees for pointing out this issue. We have clarified this in the revised text and added more details on methods of behavioral test and analysis. Also, Figure 3 annotations have been modified to provide more accurate descriptions of the results. We have also provided more details on behavioral assays in the Methods section.

The reason we used gut contraction time as a measurement in Figure 3 was that larvae with both PDF and HGN1 neurons silenced showed very little gut movements. This made it hard to quantify the gut contraction by measuring defecation intervals. We have made this point more clearly in the revised text.

*Gut peristalsis and defecation rate may vary depending on the larval stage. The authors seem to do all their experiments in L3 larva. Were these early, post-feeding, post critical weight or wandering L3s? Quantifications would need to be confined to one of these substages or data should be provided showing no differences between substages. Again there is a need for precise details of the behavioural assay and data*.

Throughout our experiment, we used larvae 96 hrs AEL. They are at active feeding stage. We have spelled it out in our revised text.

*Loss and gain-of-function phenotypes of PDF and RN2 neurons. Loss-of-function experiments were carried out constitutively throughout embryonic and larval development using Pdf-Gal4 and RN2-Gal4. Both drivers are expressed in other neurons that, when inactivated, could affect behaviours (e.g. feeding, circadian) that could indirectly impact gut movements (Thus, getting these experiments done at cellular resolution is important). Importantly, PDF has been shown to regulate tracheal branching in the hindgut, which could also affect muscle contractions (Linneweber et al Cell 2014). In the GoF muscle physiology experiments, no “raw” data has been provided for PDF neurons, and different experiments seem to have been conducted for the two neuronal populations. Only one control seems to have been used for the ChR2 experiments - several are normally required (+/ - all-trans retinal, Gal4 and UAS parental controls). More details would also have to be provided about quantifications (how are the effects of tissue compression on fluorescence controlled? The hindgut diameter should become smaller. How would the ROI remain the same if the gut is actively contracting?)*.

Loss-of-function experiments were carried out with reagents that are currently available. PDF-Gal4 was the Gal4 line with the most restricted expression pattern for those neurons. Although we weren’t able to reach single cell resolution, our data support the notion that PDF and HGN1 neurons in the VNC are important for larval defecation. We have spelled out the caveats concerning the broader expression patterns of Gal4 lines in Discussion.

We have tried to study the connectivity specificity between PDF neurons and hindgut muscles but failed to do so due to the special structure of hindguts. They only have a very thin layer of muscles, which tend to be more vigorously contracting even in a fillet preparation. We were unable to achieve stable intracellular recording of hindguts as we have done for anus sphincter muscles. However, we managed to activate PDF via ChR2 and induce anal muscle contraction, similar to what was done in HGN1>ChR2 larvae (Figure 2). Recording of anal sphincter in PDF>ChR2 larvae revealed that PDF neuron activation gave rise to anal sphincter muscle EJPs (new Figure 2). These results suggested PDF neurons also regulate anal sphincter movement, probably via HGN1 neurons.

We have added the data of control experiments (+/- all-trans retinal, Gal4 and UAS parental controls) to the figures.

We have also provided more details about the quantification of this behavior.

*The authors state that both PDF and RN2 neurons are motor neurons, based on comparisons with the gut innervation seen using VGLUT-Gal4. This is unconvincing. The authors should show the positions of the PDF, RN2 and OK371 axons on the hindgut muscles side by side, and use a broad neuronal marker like 22c10 to co-label the major nerves*. *What functional data shows a motor rather than neuromodulatory role for these neurons, especially for PDF?*

We have performed additional experiments to address this question:

For PDF neurons: We carried out anti-vGlut staining on hindgut and found the PDF axons are vGlut positive (in new Figure 1). We also used vGlut-Gal4 to label motor neurons and found they labeled the same group of axons that are marked by PDF-Gal4 (in new Figure 1—figure supplement 2). These data suggest that PDF neurons are likely motor neurons.

For HGN1 neurons: We have found the HGN1 axons over anal sphincter are vGlut positive (in new Figure 1). Together with the functional study of connection between HGN1 neurons and anus sphincter muscles, these data suggest that HGN1 neurons are motor neurons.

Additionally, we have performed imaging and whole-cell recording of HGN1 neurons in dissected VNC, which were devoid of sensory inputs. We observed a robust rhythmic activity for HGN1 neurons. HGN1 neurons also exhibited burst firing patterns, which is a feature for fly motor neurons (in new Figure 4—figure supplement 1).

These new data have been integrated into our revised manuscript.

*The role of RN2 (see above too) is unclear. Activation of RN2 by ChR2 causes EJPs in the anal sphincter. This is the only direct evidence for the proposed role of RN2 as exciting the anal sphincter muscles. Other evidence suggests a more complex situation, ablation of RN2 does not prevent sphincter opening but does act additively with ablation of PDF on hindgut contraction period (*Figure 3*, also see comments re*
Figure 3
*below). So does RN2 activation, by ChR2, alone have an effect on hind gut peristalsis? This data should be added to*
Figure 2.

We have monitored the contraction of hindgut while stimulating HGN1 neurons using a setup similar to the one we used to observe the contraction of anus sphincter. We found activation of HGN1 neurons had no effect on hindgut movement (in new Figure 2).

Regarding the question why “ablation of HGN1 does not prevent sphincter opening but does act additively with ablation of PDF on hindgut contraction period”, one speculation is that the remaining PDF neurons can still trigger hindgut contractions, which generate pressure and push the feces along the intestine to induce passive sphincter opening. We have included possible explanation in the revised text.

*The sequential firing is very neat. There are several experimental predictions that come from this. One is that activation of PDF by ChR2 should induce anal muscle contraction (via RN2), data should therefore be included in*
Figure 2
*on the anal sphincter muscle contraction when PDF is activated by ChR2. Similarly activation of PDF by ChR2 would be expected to give rise to anal sphincter EJPs. Another expectation is that inactivation of PDF may result in no Ca2+ signals in RN2, this could be tested by combining available reagents*. *Finally would it also be possible to test whether activation of PDF using ChR2 led to calcium signals in RN2?*

We used PDF-Gal4 to drive ChR2 in PDF neurons and stimulated these neurons with blue light. We found that activation of PDF via ChR2 induced anal muscle contraction, similar to what was observed in HGN1>ChR2 larvae (new Figure 2). Recording of anal sphincter in PDF>ChR2 larvae revealed that PDF neuron activation gave rise to anal sphincter muscle EJPS (new Figure 2).

*The authors seem to have used HGN1-Gal4 to label the RN2 neurons. As described in Cognigni et al, HGN1-Gal4 is a particular line (the D insertion) resulting from mobilizing the initial RN2-Gal4 (Landgraf et al). This is an important distinction because other RN2-Gal4 lines are either expressed more broadly, or are not expressed in those hindgut-innervating neurons and HGN1-Gal4 should be used in the text as this will allow other workers to connect the two studies*.

We have replaced RN2 with HGN1 throughout the manuscripts. We agree with the referees that the latter terminology will help the readers to connect our study with the previous work in the adult system.

The data showing that PDF neurons are presynaptic to RN2 is not very convincing (especially the sytGFP/Denmark figure) - both neurons could be putting out dendrites in the same VNC region in order to respond to the same presynaptic partner.

We have modified the text in the revised manuscript accordingly. For now we don’t have enough evidence to show PDF neurons are direct pre-synaptic partner of HGN1 neurons even though their aborizations overlap intensely. Based on our functional study (dual-imaging of the two groups of neurons), they are likely two components within the defecation motor circuitry where PDF neurons fire shortly before the firing of HGN1 neurons.

*The sensory role of the ASN is potentially interesting, but there is no data linking that particular ppk neuron (as opposed to other ppk+ neurons) with the hindgut muscle contraction phenotypes and/or defecation control. The authors could use the elegant approach previously used by this lab (Yang et al Neuron 2009,*
Figure 3*) to strengthen this link*.

The method suggested by the referees would provide one test as to whether ASN itself is important for the defecation. With the reagents currently available, the experiment we could do is to use 19-12-Gal4/UAS-Kir to silence all class III da neurons and PPK-Gal80 to suppress the silencing effect in ASN (19-12-Gal4 and PPK-Gal4 only overlap in the ASN). However, this would generate indirect evidence.

We used an alternative and more direct approach instead. We labeled all PPK neurons with PPK-Gal4 and used 2-photon laser to specifically ablate the ASN soma near the anal slit. Then we imaged the HGN1 neurons to see if ablation of ASN altered the activity of HGN1 neurons. Indeed, we found alteration of the periodicity of rhythmic firing of HGN1 neurons (new Figure 5).

The result supports the conclusion that ASN (a particular PPK neuron) but not other PPK neurons modulates motor control of gut movements.

*ASN is identified as sensory neuron providing mechanosensory feedback from the anus. ASN shows periodic Ca2+ oscillations. These correlate to the opening of the anus and activity in PDF and RN2, however no functional test of the hypothesis that ASN activation lies downstream of PDF, RN2 and anal opening is shown. One experiment would be to show that inactivation of PDF results in a loss of Ca2+ signals in ASN. A more positive experiment would be to show that activation of RN2 (using ChR2 or dTRPA1) in turn activates ASN. The feedback experiments use a very indirect test of feedback onto PDF and/or RN2, EJPs in the anal sphincter muscles. This may be acting through ASN-PDF-RN2-anal sphincter muscle series. However there are other explanations, in particular it does not rule out other routes of ASN feedback onto the anal sphincter muscles. It would be much better to test the effect of ASN activity more directly on PDF. For example do NOMPC mutants have changes in PDF Ca2+ signals, which are alleviated by NOMPC rescue in ASN? Does ablation of ASN change PDF calcium signals? Even better would be activation of ASN and direct measurement of signals in PDF. This would add significant weight to the argument that ASN is a functional modify of PDF properties*.

We agree that recording from gut muscles while stimulating ASN didn’t provide direct evidence that ASN feedback to PDF and HGN1 neurons. Hence we carried out experiments using new strategies in order to show that ASN does send feedback to PDF and HGN1 when it receives mechanical stimulus. We imaged PDF neurons and HGN1 neurons with GCaMP5 while inserting a tapered glass probe to the anus slit and advanced it to split the anus sphincter, which mimicked anus opening during defecation. We found that PDF and HGN1 neurons responded to this stimulation (new Figure 5). As a control experiment to show that this probe stimulation activates ASN but not other PPK neurons, we imaged PPK neurons’ axonal projection in the VNC. Stimulation with a probe on the anus induced an asymmetric Ca^2+^ increase in the VNC (new Figure 5—figure supplement 2), which is consistent with asymmetric projection of ASN axons to the VNC.

This result, together with the previous GRASP experiments between ASN and PDF neurons, support our model that ASN provide feedback to motor neurons.